# Cognitive Impairment in End Stage Renal Disease Patients Undergoing Hemodialysis: Markers and Risk Factors

**DOI:** 10.3390/ijerph19042389

**Published:** 2022-02-18

**Authors:** Piotr Olczyk, Mariusz Kusztal, Tomasz Gołębiowski, Krzysztof Letachowicz, Magdalena Krajewska

**Affiliations:** Department of Nephrology and Transplantation Medicine, Wroclaw Medical University, Borowska 213, 50-556 Wrocław, Poland; ol.piotr1994@gmail.com (P.O.); tomasz.golebiowski@umw.edu.pl (T.G.); krzysztof.letachowicz@umw.edu.pl (K.L.); magdalena.krajewska@umw.edu.pl (M.K.)

**Keywords:** renal replacement therapy, cognition, dementia, cardiovascular, dialysis, cerebral blood flow, brain saturation

## Abstract

(1) Background: Cognitive impairment (CI) is common in chronic kidney disease (CKD) and patients treated with hemodialysis. (2) Methods: The systematic review was prepared following the PRISMA statement (2013). The biomedical electronic databases MEDLINE and SCOPUS were searched. (3) Results: out of 1093 studies, only 30, which met problem and population criteria, were included in this review. The risk factors for CI can be divided into three groups: traditional risk factors (present in the general population), factors related to dialysis sessions, and nontraditional risk factors occurring more frequently in the HD group. (4) Conclusions: the methods of counteracting CI effective in the general population should also be effective in HD patients. However, there is a need to develop unique anti-CI approaches targeting specific HD risk factors, i.e., modified hemodialysis parameters stabilizing cerebral saturation and blood flow.

## 1. Introduction

The incidence of end-stage renal disease (ESRD) requiring dialysis continued to rise and reached 554,038 in 2018 in the US. In 2018, 124,500 new ESRD cases were registered, compared to 131,636 in the previous year. The number of dialysis patients on the kidney transplant waiting list as of 31 December 2017, was 75,745 applicants, 85% of whom were waiting for their first kidney transplant. Spending on ESRD patients was USD 35.9 billion, which is 7.2% of all claims paid by Medicare on the service fee system [1]. This group is exposed to many hemodialysis side effects, such as progressive cardiovascular damage, calcium-phosphorus disorders, accelerated dementia, and cognitive decline. A significant deterioration of cognitive function in hemodialysis patients is also observed in non-elderly patients in numerous studies [2,3], which was also identified as an independent risk factor increasing mortality [4]. Cognitive impairment is common in chronic kidney disease and in patients treated with dialysis. Patients treated with hemodialysis perform worse than the general population on tests of global cognition, attention and orientation, concept formation and reasoning, construction and motor performance, executive functioning, language, and memory [2]. During hemodialysis, there are significant changes in the circulatory system. The reason for this is the loss of water, both in the ultrafiltration process and from migration to tissues from blood vessels. This leads to a decrease in blood volume and an increase in its density and viscosity. Peripheral resistance is also increasing. All of these changes cause impairment of tissue blood supply, including the central nervous system [5]. A systematic review of risk factors in this group of patients was prepared, taking into account the importance of cognitive impairment in hemodialysis patients.

## 2. Materials and Methods

The systematic review was created using the “Preferred Reporting Items for Systematic Reviews and Meta-Analysis” (PRISMA, 2013). Furthermore, with the assistance of a professional health science librarian at the authors’ university, biomedical electronic databases such as SCOPUS and MEDLINE were searched. Figure 1 depicts a flow diagram outlining the literature search strategies. To identify relevant search terms, preliminary database searches were performed with the search terms (cognitive or cognition or cognitive decline or cognitive impairment) AND (hemodialysis). The systematic review includes works published 10 years prior to 30 April 2021. The PICO algorithm was used to establish the criteria for including papers in the review. The following standards were devised: P (population)—adult hemodialysis patient with cognitive impairment; I (Intervention)—there is no intervention; C (Comparison)—with a healthy population or a hemodialysis patient who does not have cognitive impairment; O (Outcome)—the statistical relationship that exists between cognitive decline and a specific factor. The terms “Alzheimer’s disease” and “dementia” were not included as search terms because the current study focused on cognitive impairment in hemodialysis patients, which is mostly a mild cognitive disorder and is thought to differ from dementia or Alzheimer’s disease in terms of risk factors or correlates. Excluded studies met the following criteria: (1) animal model studies; (2) studies with insufficient statistical data; (3) review studies; (4) abstract-only studies; (5) qualitative studies; (6) case reports; or (7) gray literature studies (theses, conference proceedings, grant or technical reports). Two scientists searched the databases, and an experienced work supervisor clarified any doubts about whether a particular piece of work was included in the review.

## 3. Results

Findings from identified studies were classified as markers of the inflammatory process and cell damage, markers of uremic toxins, disorders associated with ESRD, systemic cardiovascular risk factors, fluctuations in cerebral blood flow associated with dialysis sessions, and unclassified lifestyle and socioeconomic factors.

### 3.1. Markers Related to the Inflammatory Process and Cell Damage

Table 1 displays markers associated with the inflammatory process or cell damage in dialysis patients.

S100B is one of the markers associated with CI in HD patients. It is a calcium-binding protein that is found in Schwann cells, astrocytes, and glial cells. Multiple logistic regression analyses in the cited study revealed that serum S100B level was a statistically significant independent predictor of CI [6].

The OPG/RANK/RANKL system is important in skeletal health and bone biology, as well as in other tissues such as the lungs, brain, and heart [7,8]. It is also linked to the pathophysiology of cardiovascular disorders with a vascular component, such as diabetes and atherosclerosis [9]. In HD patients, a link was discovered between serum log-transformed RANKL levels and cognitive function tests (MoCA and CASI) [10].

Many studies conducted in general population suggest an association between cognitive functions and markers of endothelial damage [11]. Endothelial cells express, among others, intracellular adhesion molecule 1 (ICAM-1), vascular cell adhesion molecule 1 (VCAM1) or Syndecan-1. Syndecan-1, ICAM-1, and angiopoietin-2 (AGPT2) were found to be associated with CI in a study by Freire de Medeiros [12]. However, AGPT2 was found to have the strongest association with cognitive function (r = −0.316, *p* = 0.001). In stressed endothelial cells, angiopoietin-2 acts as an autocrine protective factor. High AGPT2 levels in CKD patients (particularly in G5) are associated with vascular stress due to abnormal fluid status and play a role in fluid distribution and accumulation.

Because CI is associated with cardiovascular mortality and vascular factors play a significant role in neurodegenerative disease, more research with endothelial damage markers is warranted.

The following study looked at the relationship between brain-derived neurotrophic factor (BDNF), inflammatory cytokines (TNF-, IL-6), fibroblast growth factor (FGF)-23 and its co-receptor -klotho, and platelet (PLT) count in HD patients. TNF-, BDNF, and blood PLT levels, as well as IL-6, were found to have statistically significant relationships [13]. BDNF is a protein that is distributed and synthesized in the nervous system (CNS). Furthermore, it is important in the differentiation, survival, and growth of neurons during CNS development [14,15]. Furthermore, inflammatory cytokines play an important role in the pathogenesis of hemodialysis-related side effects associated with brain diseases [16,17,18]. This study found that IL-6 and TNF- levels increased significantly in HD patients. Their levels differed significantly between the control and MCD groups.

According to the research, insulin-like growth factor-1 (IGF-1) can be used as a modern biomarker to assess cognitive functioning in HD patients. The data revealed a statistically significant difference in IGF-1 levels among all three groups (severe CI, moderate CI, and no CI) [19]. Furthermore, increased serum IGF-1 levels may lower the risk of developing dementia, as IGF-1 is involved in the removal of beta amyloid from the brain. As a result, it can be assigned a beneficial role in cognitive function improvement [20].

The next marker under consideration is fibroblast growth factor-23 (FGF-23), whose level is elevated in HD patients and is linked to left ventricular hypertrophy and increased mortality [21,22]. Furthermore, the presence of FGF-23 in the brain has been demonstrated [23,24]. In another study, elevated levels of FGF23 were linked to memory deterioration as measured by composite memory scores. This suggests that in HD patients, this marker may contribute to CI [25].

### 3.2. Markers Related to Uremic Toxins

Among uremic markers, serum uric acid (SUA) has been linked to CI. Some studies found that a higher level of SUA was associated with poorer cognitive function [26,27], while others found that it was beneficial [28,29], which could be related to uric acid antioxidant and oxidant function in neurons [30]. A study on chronic hemodialysis patients found a negative correlation between SUA and MMSE scores (r = −0.307, *p* = 0.014), which was independent of clinical and demographic factors [31].

The next marker, indole-3-acetic acid (IAA), is a uremic solute that is protein-bound. IAA was found to be responsible for the activation of the AhR/p38MAPK/NF-B inflammatory pathway in human endothelial cells, which resulted in increased ROS production, stimulation of cyclooxygenase-2, and tissue factor expression [32]. It is also known that oxidative stress and nervous system inflammation contribute to neurodegeneration [33]. Furthermore, both studies [34,35] discovered a link between IAA and cognitive impairment in HD patients.

Another study used metabolic profiling to identify uremic metabolites associated with impaired executive function in two groups of patients receiving maintenance dialysis. Executive function impairment has been linked to four metabolites: prolyl-hydroxyproline, phenylacetylglutamine, 4-hydroxyphenylacetate, and hippurate [36].

Hyponatremia is another marker associated with CI in HD patients, according to the literature. However, it is not a common abnormality associated with renal failure. In this difficult patient population, it is more frequently associated with other comorbidities (e.g., heart failure, dementia disorders). Significant correlations were discovered between CI, depression symptoms, and serum sodium levels. Furthermore, such differences have been observed in hyponatraemia of any severity (moderate to severe) [37]. The following study discovered a link between impaired functional status and mild chronic hyponatremia [38]. One possible explanation is that uremia can cause neuronal damage, which can lead to cognitive decline and lower serum sodium levels in HD patients [39]. Another theory is that hyponatremia alters the amino acids in the brain [40], which may be related to cognitive function [41].

### 3.3. Disorders Associated with ESRD

Several studies have confirmed the importance of vitamin D in cognitive disorders (see Table 1). Patients with renal insufficiency are known to have low vitamin D levels. Furthermore, vitamin D has been shown to play a role in neuroprotection via glial cell-derived neurotrophic factor, nitric oxide synthase, and nerve growth factor. Vitamin D appears to promote neuron survival, increase antioxidant activity, improve oxidative stress-induced mitochondrial dysfunction, and reduce the effect of excitatory neurotoxins. Vitamin D also reduces amyloid precursor transcription, which prevents amyloid-beta accumulation. As a result, it is understood that a lack of Vitamin D reduces neurological function and thus the CI [42,43]. The first study found that patients with cognitive impairment were more likely to be African Americans, women, diabetics, and patients who have been on dialysis for a longer period of time. Higher levels of 25 (OH) D correlated with results in executive function tests for each SD higher level of 25 (OH) D. In the case of memory assessment tests, there was no correlation [44].

Anemia is another factor to consider, as it increases the cerebral oxygen extraction fraction in HD patients and may impair cerebral vasodilator capacity [45]. Cognitive impairment is common in ESRD and is associated with poor outcomes, and anemia in these patients can lead to cerebral ischemia, cognitive impairment, and dementia [43]. The study confirms that anemia can cause CI in patients with HD. The first study found that anemia is linked to mild to moderate cognitive impairment in people with Parkinson’s disease. The results of cognitive function tests improved as the hemoglobin level increased. Furthermore, blood flow increased in the middle cerebral artery (MCA). The greatest improvement was seen in stage 3 (Hb 11.5–12.5 g/dL, 7.14–7.76 mmol/L) compared to stage 2 (Hb 10–11.5 g/dL, 6.21–7.14 mmol/L). In the case of MCA, Hb values of 11.5–12.5 g/dL were associated with the greatest improvement in cognitive function and cerebral circulation in transcranial Doppler (TCD) testing [46]. Another retrospective study examined data from 43, 906 adult HD patients using Cox’s hazard ratio and regression models. EPO supplementation was associated with a 39% lower risk of developing systemic dementia compared to patients who did not receive EPO supplementation. The risk of incurable dementia (UnD) and vascular dementia (VaD) was also lower in the EPO group. Patients who received both EPO and iron preparations reaped additional benefits [47].

### 3.4. Cognitive Function and Systemic Cardiovascular Risk Factors

Table 1 shows studies that link CI to cardiovascular risk factors. Many studies have found that pulse wave velocity (PWV) and ankle-brachial index (ABI) are important factors associated with cognitive impairment in dialysis patients (ABI). These parameters have been validated as tools for assessing arterial health as measured by arterial stiffness. Peripheral arterial disease (PAD) has been shown to increase the risk of both cardiovascular disease [48] and cognitive impairment in the general population [49]. The studies found suggest that such a relationship exists in the group of HD patients as well. The first scientific study found that having a high PWV or a low ABI is associated with poor cognitive function in HD patients [50]. Other studies [51] support the link between PWV and cognitive impairment in hemodialysis patients. Orthostatic pressure reduction is another cardiovascular factor linked to cognitive performance. According to research, there is a link between CKD and the impairment of orthostatic pressure stabilization, specifically the orthostatic systolic function [52,53]. It is well understood that impaired renal function reduces baroreceptor sensitivity and causes autonomic dysfunction, which increases the risk of orthostatic hypotension [54,55]. According to the findings of the study, an excessive reduction in orthostatic pressure in HD patients causes memory impairment [56]. The common carotid pulsation is an indicator of cerebral microvascular microangiopathy (CCAPI). Furthermore, studies show that CCAPI correlates with cognitive function in HD patients with no history of stroke or dementia [57]. Another factor to consider is left ventricular function. CKD increases the risk of developing left ventricular hypertrophy at a young age. In the early stages of HD, 70–80 percent of patients have left ventricular hypertrophy [58,59]. Furthermore, chronic hemodialysis reduces cerebral blood flow, which may exacerbate the effects of low LVEF [60]. According to research, a mildly reduced LVEF correlates with cognitive impairment [61].

### 3.5. Fluctuations in Cerebral Blood Flow and Cognitive Function

Table 1 summarizes findings from studies that link CI to abnormal cerebral blood flow. It has been proposed that hemodialysis causes brain damage. It is associated with recurrent hemodynamic changes, specifically a decrease in cerebral intradialytic perfusion. Dialysis factors such as ultrafiltration volume or intradialytic hypotension could be one cause of this phenomenon [62,63]. White matter hyperintensity (WMH) is a validated marker of small vessel diseases and changes in brain structure [64]. The first of the studies presented examines the relationship between cognitive functions in HD patients and mean flow velocity in the cerebral arteries (MFV). The volume of ultrafiltrates has been linked to a decrease in MFV during HD. Furthermore, it was discovered that the decline in cognitive functions (executive functions, global functions, and verbal fluency) during dialysis was related to the decline in MFV. Furthermore, 73 HD patients were re-examined after a year. Reduced global and executive function in these patients was significantly related to the percentage of MFV decline and progression of WMH burden [65].

Another study used quantitative sensitivity mapping to assess the relationship between the results of neuropsychological tests and clinical factors with non-invasive assessment of changes in regional cerebral venous blood saturation (rSO _2_) in HD patients (QSM). In HD patients, the SO_2_ of bilateral thalamocortical, cortical, internal and basal, septal, and basal veins was lower than in the healthy control group (HC). The Montreal Cognitive Assessment (MoCA) and Mini-Mental State Examination (MMSE) scores were both lower in HD patients, and the MoCA scores correlated with SO_2_ levels in the brain’s left internal vein. Clinical parameters such as iron levels, hematocrit, blood pressure before and after dialysis, and glucose have been shown in studies to be independent risk factors for cerebral rSO_2_. According to research, cerebral rSO_2_ can be considered a risk factor for cognitive disorders [66]. Another two studies used the INVOS 5100c system to assess regional saturation of the frontal lobes. The first found that HD patients with cognitive impairment had lower rSO_2_ levels in the brain than patients with normal cognitive functions [67]. The second study found that changes in rSO_2_ levels in the brain are significantly related to hemoglobin levels, pulse rate, and serum albumin levels [68].

All of these studies indicate that a better understanding of the causes of cerebral ischemic stroke in HD patients could help to prevent cognitive decline in this population.

**Table 1 ijerph-19-02389-t001:** Grouped markers and risk factors of cognitive impairment in hemodialysis patients.

Correlates/Surrogates of Cognitive Impairment	Studied Population	Cognitive Impairment Assessment	Ref.
Markers Related to the Inflammatory Process and Cell Damage			
Neurobiomarker S100 calcium binding protein B (S100B); S100B level was independent predictor of CI (cut-off values for predicting CI was 36.1 pg/mL).	30 HD *	MMSE	[6]
Bone turnover marker RANKL (Receptor activator of nuclear factor-kappa B ligand) level linked with better cognitive function. MoCA (β = 1.14, 95% CI 0.17 to 2.11) and CASI (β = 3.06, 95% CI 0.24 to 5.88).	251 HD 37 HC **	MoCA scale Cognitive Abilities Screening Instrument (CASI)	[10]
Endothelium-related biomarkers: syndecan-1, intercellular adhesion molecule-1 (ICAM-1), and angiopoietin-2 (AGPT2) correlated with better CI.	216 HD	Cambridge Cognitive Examination MMSE	[12]
Brain-derived neurotrophic factor (BDNF) and platelets count correlated with cognitive test scores.	58 HD 20 HC	MMSE MoCA scale	[13]
Insulin-like growth factor-1 (IGF-1) low levels is risk factor for severe CI and dementia.	93 HD	MMSE	[19]
FGF-23 linked with worse performance on a composite memory score; FGF-23 was independently associated with a lower memory score.	263 HD	Wechsler Memory Scale-III, Word List Learning Subtest, Wechsler Adult Intelligence Scale-III, Block Design and Digit Symbol-Coding Subtests, Trail Making Tests A and B	[25]
Uremic Toxins			
Uric acid level showed negative correlation with MMSE score (r = −0.307, *p* = 0.014).	180 HD	MMSE	[31]
Protein-bound uremic solute—indole-3-acetic acid (IAA) serum level was associated with a poor MMSE (β = −0.90) and a poor CASI (β = −3.29).	230 HD	MMSE MoCA CASI	[34]
Circulating free indoxyl sulfate levels were negatively associated with the MMSE scores (β = −0.62) and the CASI scores (β = −1.97).	260 HD	MMSE CASI	[35]
4-hydroxyphenylacetate (RR = 1.16), hippurane (RR = 1.24), phenylacetylglutamine (RR = 1.39), prolyl-hydroxyproline (RR = 1.20) showed association with CI scores.	141 HD 180 HC	Trail Making Test Part B Digit Symbol Substitution Test	[36]
Hyponatremia correlated with symptoms of depression.	200 HD	Patient Health Questionnaire Perceived Deficit Questionnaire-5	[37]
Plasma phosphorus level (>6 mg/dL, *p* = 0.034), inadequate dialysis dose (Kt/V < 1, 2, *p* = 0.023) and hyponatremia (Na < 135 mEq/L, *p* = 0.001) infuenced poor executive and functional status.	56 HD	Modified Mini-Mental State (3MS) Trail Making Test A and B	[38]
Disorders Associated with ESRD			
Vitamin D—25(OH)D levels correlated with executive functions (β = 0.16; *p* < 0.05) but no with memory assessment tests.	255 HD	MMSE, Wechsler Memory Scale-III (WMS-III), Word List Learning Subtest Wechsler Adult Intelligence Scale-III (WAIS-III) Block Design and Digit Symbol-Coding subtests Trail Making Test A and B	[44]
Anemia correlated with CI (increase in Hb values improved cognitive functions); improvement in Hb (*p* < 0.05) correlated with cerebral artery blood flow.	120 HD	MMSE	[46]
Patients receiving EPO had a 39% lower risk of general dementia than those in the non-EPO group. The risk of dementia was further reduced in HD patients with EPO treatment in combination with iron.	43, 906 HD	Clinical data	[47]
Systemic cardiovascular risk factors			
Ankle-brachial index ABI < 0.9 showed association with the MoCA score (β = 0.62, *p* = 0.011) and the CASI score (β = 1.43, *p* = 0.026). Arterial stiffness surrogate—baPWV showed negative correlation with CASI (β = −0.70, *p* = 0.009).	136 HD	MoCA CASI	[50]
Pulse wave velocity (PWV) values were associated with worse MMSE scores (β = −0.36, *p* = 0.001), and MiniCog scores (β = −0.26, *p* = 0.02). PWV value was significantly associated with TMTA but not with TMTB.	72 HD	MMSE, Part A (TMTA) and Part B (TMTB) Mini-Cog Test	[51]
Maximum orthostatic systolic blood pressure reduction was independently and negatively associated with short (β = −0.05, *p* = 0.029) and delayed (β = −0.05, *p* = 0.035) recall memory in dialysis patients but not in controls.	80 HD 80 HC	MoCA Auditory Verbal Learning Test (AVLT)	[56]
Common carotid artery pulsation index (CCAPI) had an independent effect on attention retention in HD patients (β = −0.36, *p* = 0.01).	37 HD 18 HC	MoCA	[57]
Left ventricle function—LVEF showed inverse association with cognitive impairment (β = 0.87, *p* = 0.022).	72 HD	MMSE	[61]
Fluctuations in cerebral blood flow			
Marker of ischaemic cerebral small-vessel disease: prevalence of white matter hyperintensities (WMH) on magnetic resonance imaging was significantly higher in HD patients than in the healthy subjects (*p* < 0.01).	179 HD 58 HC	WMH (white matter hyperintensities) on MRI is known CI risk factor in the general population	[64]
Cerebral arterial mean flow velocity (MFV) decline was correlated with the intradialytic decline in cognitive functions, including global functions, executive functions, and verbal fluency (*p* < 0.01).	97 HD	National Institute of Neurologic Disorders and Stroke-Canadian Stroke Network Neuropsychological Battery	[65]
Reduced regional cerebral venous oxygen saturation (SvO) of two bilateral cortical, thalamic, septal, internal cerebral and basal regions in HD patients was significantly lower than in HC.	54 HD 54 HC	MMSE MoCA	[66]
Cerebral exigenation (rSO2) values in HD patients was lower compared to cognitively healthy people. The relation between rSO2 and MoCA score was significant after adjustment for age and gender (*p* = 0.007).	39 HD	MoCA	[67]

* HD—group of hemodialysis patients, ** HC—group of healthy control patients, MMSE—Mini-Mental State Exam, MoCA-Montreal Cognitive Assessment.

### 3.6. Other Unclassified Factors

Table 2 shows additional socioeconomic and lifestyle factors. Higher levels of education are associated with lower risks of cognitive impairment and mortality in the general population. This relationship exists in the HD group as well. In this group of patients, educational level, post-dialysis blood pressure, and socioeconomic status are independent factors of CI. Another correlation that exists in both the HD group and the general population is a higher prevalence of CI in people who are depressed. The longer the hemodialysis vintage, the greater the risk of CI [69,70,71,72]. The following article discusses the connection between CI and sleep quality. A lower risk of CI is directly related to better sleep quality [73]. Physical activity is another important factor. The findings of this study suggest that encouraging dialysis patients to exercise may reduce their risk of developing CI [74].

## 4. Conclusions

Many biomarkers that correlate with CI in HD patients show that the pathogenesis of these changes in this group of patients is very complicated. Furthermore, the majority of the presented relationships will need to be confirmed in future studies. More research is needed in this area to identify key risk factors. Then, knowing the primary causes can aid in the design of nonpharmacological intervention studies to reduce the risk of CI in HD patients.

The risk factors for CI can be divided into three groups based on the review: traditional risk factors (present in the general population), dialysis-related risk factors, and nontraditional risk factors that occur more frequently in the HD group. Age, education, sleep quality, and depression are all traditional risk factors (ABI 0.9). Among the HD-related factors are dialysis vintage, uremic toxins (SUA, serum IAA, free IS levels, 4-hydroxyphenylacetate, phenylacetylglutamine, hippurate, prolyl-hydroxyproline), and EPO treatment quality. The final group of risk factors includes nontraditional risk factors that are more common in ESRD and hemodialysis patients. This category includes markers related to the inflammatory process and cell damage (serum levels of S100B, RANKL, ICAM-1, AGPT2, syndecan-1 HR, BDNF, TNF, interleukin-6, PLT, IGF-1, IGFBP-3), FGF-23, anemia, 25(OH)D levels, cardiovascular risk factors (PWV, CCAPI, LVEF), mean flow velocity in the cerebral arteries, or changes in regional cerebral blood (see Figure 2). On this basis, it can be concluded that the methods for combating CI that work in the general population can also work in HD patients. However, there is a need to develop novel anti-CI strategies that target specific HD risk factors, such as modified hemodialysis parameters (continuous on-line blood pressure, UF, temperature, saturation monitoring).

## Figures and Tables

**Figure 1 ijerph-19-02389-f001:**
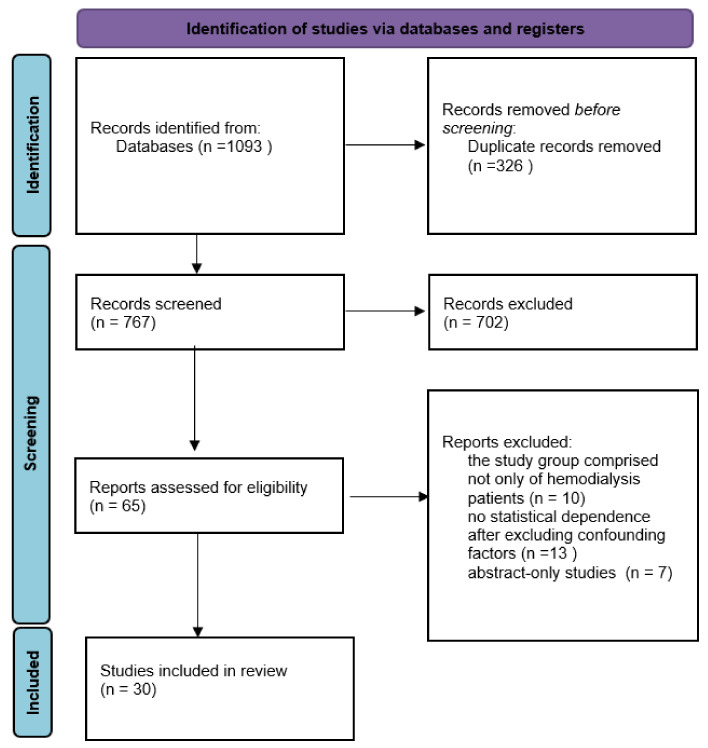
A flow diagram outlining the literature search strategies.

**Figure 2 ijerph-19-02389-f002:**
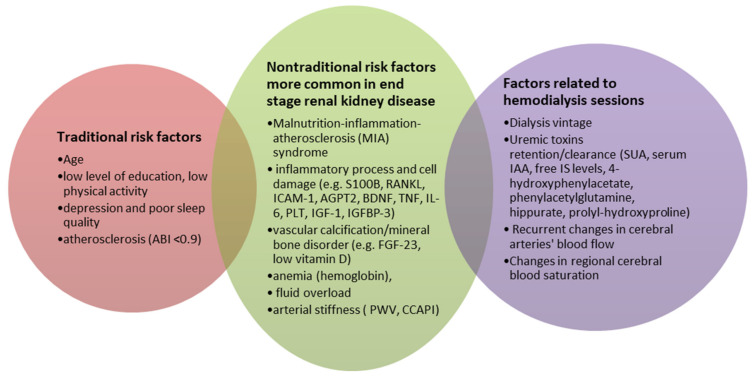
Risk factors for cognitive impairment in hemodialysis patients.

**Table 2 ijerph-19-02389-t002:** Other unclassified factors influencing cognitive functions.

Factors	Studied Population	CI Measures	Ref.
Factors correlating with CI were identified: Educational level (OR 2.234), spKt/V (OR 1.982), Post-dialysis diastolic blood (OR 1.982).	219 HD *	MoCA	[69]
Identified correlates: socio-economic status and global cognition score (χ^2^ = 81.13, df = 48, *p* = 0.002), education level and orientation (χ^2^ = 29.78, df = 8, *p* = 0.000), recall (χ^2^ = 31.7, df = 12, *p* = 0.002). A negative correlation was found between dialysis vintage (r = −0.411, *p* = 0.003), depression (r = −0.721, *p* <0.01) and cognitive function.	50 HD	MoCA Patient Health Questionnaire-9 (PHQ-9)	[70]
A positive correlations was found between cognitive function and years of education (r = 0.52, *p* ≤ 0.001), dialysis vintage (r = 0.26, *p* ≤ 0.001).	99 HD	Addenbrooke’s Cognitive Examination-Revised (ACE-R)	[71]
Educational level (odd ratio = 0.564, *p* = 0.031), anemia (odd ratio = 0.743; *p* = 0.046) assiociated with cognitive functions.	108 HD	MMSE	[72]
Sleep quality (OR 10.709 *p* = 0.002) independently associated with CI.	106 HD	British Columbia Cognitive Complaints Inventory	[73]
Less physically active patients assiociated with CI.	102 HD	MMSE	[74]

* HD—group of hemodialysis patients; MMSE—Mini-Mental State Exam, MoCA—Montreal Cognitive Assessment.

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
