# Peer review of "Cognitive Impairment in End Stage Renal Disease Patients Undergoing Hemodialysis: Markers and Risk Factors"

_ijerph, 2022, doi:10.3390/ijerph19042389_

Round 1

Reviewer 1 Report

The Report represents a very interesting and well-planned review devoted to the study of risk factors and markers for cognitive impairment (CI) in hemodialysis (HD) patients.

Used databases are named, eligibility criteria are specified. It would be quite interesting to know relevant search terms, which were identified during the search process.

During the study markers and risk factors of cognitive impairment in hemodialysis patients were grouped and analysed. The approach is quite interesting and helpful for understanding the changes associated with CI in HD patients. The prepared review would help to develop a method of counteracting CI taking into account specific hemodialysis risk factors, which is quite important.

The following comments do not diminish significance of the Article:

Line 25 Probably it would be good to add information about the number of end-stage renal disease (ESRD) requiring dialysis incidences, number of new ESRD cases which occured since 2019 until 2021 time period, if possible.

Line 66 The following fragment “. .” would be better to changed to “.”

Line 69 It would be better to correct the flow diagram, Figure 1, as text in the frames partly disappeared. And diagram includes star symbols, which meaning probably should be explained.

Line 106 Probably it would be better to place Table 1 before the paragraph ‘3.1 Markers Related to the Inflammatory Process and Cell Damage’, because the table contains information which is also described in several paragraphs later.

Line 109 Table 1, headlines, the following fragment “Studied pupulation…” would be better to change to “Studied population…”

Lines 202, 203 Probably it would be good to add Hb values in SI units also.

Line 232 Probably it would be better to check the phrase “dementia, stroke or dementia”.

Line 284 Table 2, factor name “Sleep quality (OR 10.709 p =0.002) idependently assiociated with CI” probably would be better to change to “Sleep quality (OR 10.709 p =0.002) independently associated with CI”.

Line 308 Figure 2, column Traditional risk factors, the following fragment “low level of ducation” probably should be changed to “low level of education”.

Probably it would be better to add Conclusion at the end of the Paper as this part is mentioned in the Abstract.

It would be better to check the references description in accordance with IJERPH requirements.

Author Response

We are gratefully for deep insight and suggestions how to improve  the manuscript. Please accept our responses and explanations given below.  Thank you so much for your time and expertise.

Reviewer #1

1.It would be quite interesting to know relevant search terms, which were identified during the search process.

   Answer: For search process of databases we used following  terms : “cognitive or cognition or cognitive decline or cognitive impairment” AND “hemodialysis”. It is displayed in lines 50-51.

2.Line 25 Probably it would be good to add information about the number of end-stage renal disease (ESRD) requiring dialysis incidences, number of new ESRD cases which occured since 2019 until 2021 time period, if possible.

3.Line 66 The following fragment “. .” would be better to changed to “.”

Line 69 It would be better to correct the flow diagram, Figure 1, as text in the frames partly disappeared. And diagram includes star symbols, which meaning probably should be explained.

Line 106 Probably it would be better to place Table 1 before the paragraph ‘3.1 Markers Related to the Inflammatory Process and Cell Damage’, because the table contains information which is also described in several paragraphs later.

Line 109 Table 1, headlines, the following fragment “Studied pupulation…” would be better to change to “Studied population…”

Lines 202, 203 Probably it would be good to add Hb values in SI units also.

Line 232 Probably it would be better to check the phrase “dementia, stroke or dementia”.

Line 284 Table 2, factor name “Sleep quality (OR 10.709 p =0.002) idependently assiociated with CI” probably would be better to change to “Sleep quality (OR 10.709 p =0.002) independently associated with CI”.

Line 308 Figure 2, column Traditional risk factors, the following fragment “low level of ducation” probably should be changed to “low level of education”.

Probably it would be better to add Conclusion at the end of the Paper as this part is mentioned in the Abstract.

It would be better to check the references description in accordance with IJERPH requirements.

Answer: All your  suggestions has been incorporated and errors corrected. Thank you for deep insight and  remarks. 

Reviewer 2 Report

This review paper has discussed the risk factors for cognitive impairment in kidney disease patients based on 1093 studies. Here are some questions I have.

  1. On page 1, from line 35 to line 41. This statement needs references to support it. Please add them.
  2. Page 2, line 53. Is "April 30, 2021," a typo?
  3. Figure 1 is distorted, please revise them and use a .tif image.
  4. Results on page 6 can be better performed by using a bar chart with significance labels.

This manuscript has been classified as a brief report. However, a lot of details are necessary to be clarified. Deep thinking and explanations are missing, for example, on Page 6, line 120. The author mentioned a study that investigated the association of endothelial markers in HD patients. Cognitive function was correlated with all endothelial-related biomarkers except VCAM-1. However, the strongest association was observed with AGPT2. Basically, the author only summarized the phenomenon but did not provide any reason.

I understand there is a length limit for the brief report, but only summarizing the result makes the manuscript superficial (lake of solid content). This is the reason I believe the author should make some significant changes.

Author Response

Dear Reviewers and Editor,

We are gratefully for deep insight and suggestions how to improve  the manuscript. Please accept our responses and explanations given below.  Thank you so much for your time and expertise.

This review paper has discussed the risk factors for cognitive impairment in kidney disease patients based on 1093 studies. Here are some questions I have.

  1. On page 1, from line 35 to line 41. This statement needs references to support it. Please add them.

  1. Page 2, line 53. Is "April 30, 2021," a typo?

Answer: the sentence was corrected –“ The systematic review includes works published 10 years prior to April 30th, 2021”

  1. Figure 1 is distorted, please revise them and use a .tif image.
  2. Results on page 6 can be better performed by using a bar chart with significance labels.

Answer: Thank you for this remarks. This has been corrected.

5.This manuscript has been classified as a brief report. However, a lot of details are necessary to be clarified. Deep thinking and explanations are missing, for example, on Page 6, line 120. The author mentioned a study that investigated the association of endothelial markers in HD patients. Cognitive function was correlated with all endothelial-related biomarkers except VCAM-1. However, the strongest association was observed with AGPT2. Basically, the author only summarized the phenomenon but did not provide any reason.

Answer:  Thank you for this remark. We modified this paragraph as follows. “Many studies conducted in general population suggest an association of cognitive functions with markers of endothelial damage [11]. Endothelial cells express , among others, intracellular adhesion molecule 1 (ICAM-1), vascular cell adhesion molecule 1 (VCAM1) or  Syndecan-1. Syndecan-1, ICAM-1, and angiopoietin-2 (AGPT2) were found to be associated with CI in a study by Freire de Medeiros [12]. However, AGPT2 was found to have the strongest association with cognitive function (r = -0.316, P 0.001). In stressed endothelial cells, angiopoietin-2 acts as an autocrine protective factor. High AGPT2 levels in CKD patients (particularly in G5) are associated with vascular stress due to abnormal fluid status and play a role in fluid distribution and accumulation.          Because CI is associated with cardiovascular mortality and vascular factors play a significant role in neurodegenerative disease, more research with endothelial damage markers is warranted.”                                                             

6.I understand there is a length limit for the brief report, but only summarizing the result makes the manuscript superficial (lake of solid content). This is the reason I believe the author should make some significant changes.

   Answer: Thank you for deep insight and remarks. We revised the manuscript and believe that the brief review will be helpful for other researchers.

Thank you for deep insight and  remarks. 

Reviewer 3 Report

Congratulations for your work, I have just one recommendation, to decide what terms you will use ”End-stage kidney disease (ESKD)”, like in your title, or End-stage renal disease (ESRD) like in the first paragraph, for example, but present in many other parts of the paper.

Minor English spelling is required

Author Response

Dear Reviewer and Editor,

We are gratefully for deep insight and suggestions how to improve  the manuscript. Please accept our responses and explanations given below.  Thank you so much for your time and expertise.

Congratulations for your work, I have just one recommendation, to decide what terms you will use ”End-stage kidney disease (ESKD)”, like in your title, or End-stage renal disease (ESRD) like in the first paragraph, for example, but present in many other parts of the paper.

Answer: Thank you for this remark. We decided to change title and use consequently End Stage Renal Disease.

Minor English spelling is required

   Answer: Thank you for deep insight and remarks. We made English corrections.

Thank you for deep insight and  remarks. 

Round 2

Reviewer 2 Report

After filling in more details, the manuscript looks better than the first version.

This manuscript is a resubmission of an earlier submission. The following is a list of the peer review reports and author responses from that submission.

Round 1

Reviewer 1 Report

Comments to the authors

  • There is a problem in the citation of references throughout the manuscript – there is always the indication “Error, Reference source not found”. The reviewer could not check the reported results, because the references are not numbered in the text.
  • Literature search: the reviewer did not understand what types of studies were eligible in this “systematic review”. The authors should state for example that the review included case-control studies comparing the cognitive dysfunction between HD patients and healthy controls or cross-sectional studies exploring associations of cognitive impairment with risk factors, etch.
  • Literature search: please clarify if this was a language-restricted search of the literature.
  • The authors should explore the possibility of “publication bias” in their literature search.
  • Figure 1: the authors excluded 13 studies because these studies did not show statistically significant associations in the multi-variate models. These studies are also important and need to be investigated further.
  • Results and tables: this section is too long and the structure of tables is complex and difficult to follow. For each individual study, the authors should report the design, some basic characteristics of study participants, the statistical adjustments performed in multi-variate models.
  • Results: there are some important questions that this systematic review failed to answer. For example, the authors could try to quantify the prevalence of cognitive impairment in HD patients and to explore if the prevalence of cognitive impairment is significantly higher in HD patients than in age- and gender-matched healthy controls.
  • The associations and the determinants of cognitive impairment vary considerably across studies. The question is why? Please try to explore this heterogeneity.

Author Response

Dear Reviewer,

We are gratefully for deep insight and suggestions how to improve  the manuscript. Please accept our responses and explanations given below.  Thank you so much for your time and expertise.

We fixed the problem with citation (wrong format in word procesor) and have hired a professional english editorial service to meet expectations.

  1. Literature search: the reviewer did not understand what types of studies were eligible in this “systematic review”. The authors should state for example that the review included case-control studies comparing the cognitive dysfunction between HD patients and healthy controls or cross-sectional studies exploring associations of cognitive impairment with risk factors, etch.

Answer: Detailed criteria for the inclusion of papers/studies are given “material and methods” section. We  based on the standardized PICO algorithm. There are also exclusion criteria that specify which types of works were rejected from the review.

  1. Literature search: please clarify if this was a language-restricted search of the literature.
  2. Answer: We are sorry if it was not clear. A criterion has been added to clarify that the work contains only publications in English.
  3. The authors should explore the possibility of “publication bias” in their literature search.
    Answer: Thank you for this remark. As we did not perform meta-analyses assessing intervention and only restricted systematic reviews the publication bias risk is low. We skipped data from unpublished studies and the grey literature. Our review did not generate any “strong” conclusions which could  potentially copy biased result. We focused on potentially risk factors of CI which, are not the same as in general population. In fact studies on dialysis population are relatively small what is naturally kind of bias.
  1. Figure 1: the authors excluded 13 studies because these studies did not show statistically significant associations in the multi-variate models. These studies are also important and need to be investigated further.
    Answer: We fully agree that the results of the works that did not show a statistically significant relationship in multivariable analysis are also important. However, the aim of the review was to identify already known risk factors for cognitive dysfunction in hemodialysis patients. There are many differences between studied dialysis population, mentioning only different vascular access type between countries (eg Canada vs Japan vs Italy) and physical activity time per week, surely influencing cognitive function.     We hope that organizing the knowledge of already known risk factors will contribute to further research on other possible risk factors and on ways to prevent progressive cognitive impairment in hemodialysis patients.
  1. Results and tables: this section is too long and the structure of tables is complex and difficult to follow. For each individual study, the authors should report the design, some basic characteristics of study participants, the statistical adjustments performed in multi-variate models.
    Answer: Tank you for this great remark. The table 1 has been changed and unified in revised version.
  2. Results: there are some important questions that this systematic review failed to answer. For example, the authors could try to quantify the prevalence of cognitive impairment in HD patients and to explore if the prevalence of cognitive impairment is significantly higher in HD patients than in age- and gender-matched healthy controls.
    Answer: Information was added in the introduction that cognitive disorders are more common in hemodialysis patients than in the general population. Unfortunately there is no large scale multinational epidemiology study on this topic yet.
  3. The associations and the determinants of cognitive impairment vary considerably across studies. The question is why? Please try to explore this heterogeneity.
    Answer: Thank you for this query. The tools and measures of cognitive function varies between studies. This is why we report in one column which scale was used. Different tests check different cognitive areas, and some cross-sectionally check all of them (MMSE, MoCA). Therefore, the conclusions in some works concern cognitive function in general and low score of cognitive function means just impairment. As we wanted to identified and group factors the was no space to compare tools of cognitive function measurement.

All your  suggestions has been incorporated. Thank you for remarks.

Reviewer 2 Report

 Reference citation have to be corrected, references must be numbered in order of appearance in the text. In the text there are only marks " [Error! 32
Reference source not found.]"

A conclusion that "the methods of counteracting CI that are effective in the general population should also be effective in the group of HD patients" is not exactly from a Systematic Review - why should it be so? Your conclusions sound like discussion, I would suggest to correct conclusions to more concrete and scientific.

Author Response

Dear Reviewer,

We are gratefully for deep insight and suggestions how to improve  the manuscript. Please accept our responses and explanations given below.  Thank you so much for your time and expertise.

We fixed the problem with citation (wrong format in word prosecor) and have hired professional english editing service to improve language.

A conclusion that "the methods of counteracting CI that are effective in the general population should also be effective in the group of HD patients" is not exactly from a Systematic Review - why should it be so? Your conclusions sound like discussion, I would suggest to correct conclusions to more concrete and scientific.
Answer: Thank you for this remark. We agree that this part was confusing and has been removed in revised version. We also hired professional English editing service which corrected a lot.

Reviewer 3 Report

It is a great honor to be able to review this article. Olczyk P et al. summarized the risk factors of cognitive decline in hemodialysis patients. Although this summarized review might be helpful for some physicians, there is some great concern in the manuscript. I cannot recommend this article for publication in its current form.

  1. First of all, this review should be classified as a systematic literature review rather than a systematic review. Generally, most published systematic reviews perform a meta-analysis, such as extracting the relevant data and synthesizing the extracted data. In fact, the beginning of the Method section says that this systematic review was prepared based on “Preferred Reporting Items for Systematic reviews and META-ANALYSIS.” Nonetheless, the authors just summarized the published clinical studies which meet the inclusion criteria, not analyzing any data. In order to prevent misunderstanding, the authors should clarify that this manuscript is a systematic literature review in the abstract and main text.

  1. The first impression of this manuscript is that it lists the clinical studies investigating the cognitive impairment of hemodialysis patients. The reason for this monotonousness might be that the authors do not assess the significance of each study. For readers, although the list of applicable molecules and parameters might be valuable, it is also essential which is hopeful and high-potential for clinical practices and research. To clarify it, the authors should present the evidence quality of each study, evaluating sample size, possible bias, strength, and social contributions. It is highly recommended that the authors fundamentally change the writing style and read each study cautiously.

  1. Unfortunately, each table is not easy to comprehend. Particularly, it is difficult to identify what molecules and parameters are discussed in each row.

  1. As for the Discussion section, it was difficult why the authors concluded that physical activity is vital for reducing the risk of CI. At least, there seems to be a short description of physical activity in the Results section.

         Anyway, as this manuscript is a literature review, it is questionable whether it needs a Discussion section.

  1. The reference numbers are not appropriately cited in the manuscript. Additionally, the abbreviation is also inappropriate, even in the abstract. Unfortunately, it gives the impression of a lack of respect for journals. The authors should check before the submission.

Author Response

We are gratefully for deep insight and suggestions how to improve  the manuscript. Please accept our responses and explanations given below.  Thank you so much for your time and expertise.

We fixed the problem with citation (wrong format in word procesor) and have hired professional english editing service to improve language and style.

  1. First of all, this review should be classified as a systematic literature review rather than a systematic review. Generally, most published systematic reviews perform a meta-analysis, such as extracting the relevant data and synthesizing the extracted data. In fact, the beginning of the Method section says that this systematic review was prepared based on “Preferred Reporting Items for Systematic reviews and META-ANALYSIS.” Nonetheless, the authors just summarized the published clinical studies which meet the inclusion criteria, not analyzing any data. In order to prevent misunderstanding, the authors should clarify that this manuscript is a systematic literature review in the abstract and main text.

Answer: Thank you for this remark. We  changed in appropriate sections that “systematic literature review” was performed.

You are right that meta-analysis require data quality assessment, collection and processing. Meta-analysis refers mainly to interventions evaluation. In fact  we did not perform meta-analyses assessing intervention and only restricted systematic literature review. We focused only on risk factors reported and identified in specific patient population – dialysis. Using Prisma and PICO methodology we skipped the grey literature and secondary studies/papers.

  1. The first impression of this manuscript is that it lists the clinical studies investigating the cognitive impairment of hemodialysis patients. The reason for this monotonousness might be that the authors do not assess the significance of each study. For readers, although the list of applicable molecules and parameters might be valuable, it is also essential which is hopeful and high-potential for clinical practices and research. To clarify it, the authors should present the evidence quality of each study, evaluating sample size, possible bias, strength, and social contributions. It is highly recommended that the authors fundamentally change the writing style and read each study cautiously.

Answer: Thank you for this remark. We changed table 1 into more interesting for readers. The aim of the review was to identify already known risk factors for cognitive impairment in hemodialysis patients. There are many differences between studied dialysis population by authors using different tools and dealing with different patients , mentioning only different vascular access type between countries (eg Canada vs Japan vs Italy) and physical activity time per week, surely influencing cognitive function. This is why we displayed in the table tools for cognitive impairment test and markers used to underlying main correlete/association. It is extremely difficult to weigh the strength of the influence of particular marker  and relate at the same time to the characteristics of the studied population. It often is uncomparable to other reports (natural sample size bias).  We hope that organizing the knowledge of already known risk factors will contribute to further research on other possible risk factors and design new study to prevent progressive cognitive impairment in hemodialysis patients.

  1. Unfortunately, each table is not easy to comprehend. Particularly, it is difficult to identify what molecules and parameters are discussed in each row.

 Answer: Thank you for this remark. We changed table 1 and  made unification (one instead of four). We do hope that now is more helpful for readers.

  1. As for the Discussion section, it was difficult why the authors concluded that physical activity is vital for reducing the risk of CI. At least, there seems to be a short description of physical activity in the Results section.

         Anyway, as this manuscript is a literature review, it is questionable whether it needs a Discussion section.

Answer: Thank you for this remark. We agree that this part was confusing and has been removed in revised version. We left discussion shorter.

  1. The reference numbers are not appropriately cited in the manuscript. Additionally, the abbreviation is also inappropriate, even in the abstract. Unfortunately, it gives the impression of a lack of respect for journals. The authors should check before the submission.

Answer: We fixed the problem with citation (wrong format in word procesor) and have hired professional English editing service to improve language and style.

All your  suggestions well taken and  incorporated if possible. Thank you for remarks and your time.

Round 2

Reviewer 1 Report

I have no further comments to the authors.

Reviewer 3 Report

It is a great honor to re-evaluate this manuscript. Unfortunately, the content does not seem to change fundamentally. It is impossible to recommend for acceptance. 
Although there are still many concerns in this manuscript, the journal does not give us enough time to describe cautiously. Thus, I wrote down the keywords for the author's future.

•    Why did the authors still describe PRISMA-2013?
•    IF the authors still insist that this is a systematic review and meta-analysis, the number of accessed databases is too insufficient.
•    PRISMA flow is unnecessary.
•    It is impossible to understand each Table due to its complexity.
•    The authors continuously list the molecules.
•    Discussion section is necessary for a literature review?
•    The content of main texts is just dull because these are just lists.